# The Use of Evidence-Informed Deliberative Processes for Health Benefit Package Design in Kazakhstan

**DOI:** 10.3390/ijerph191811412

**Published:** 2022-09-10

**Authors:** Wija Oortwijn, Gavin Surgey, Tanja Novakovic, Rob Baltussen, Lyazzat Kosherbayeva

**Affiliations:** 1Department for Health Evidence, Radboud University Medical Centre, 6525 GA Nijmegen, The Netherlands; 2ZEM Solutions, 11010 Belgrade, Serbia; 3Department of Health Politics and Management, School of Public Health, Asfendiyarov Kazakh National Medical University, Almaty 050000, Kazakhstan

**Keywords:** Kazakhstan, health benefit package design, evidence-informed deliberative process

## Abstract

Kazakhstan strives to obtain Universal Health Coverage (UHC) by using health technology assessment (HTA) for determining their health benefit package. This paper reports on employing evidence-informed deliberative processes (EDPs), a practical and stepwise approach to enhance legitimate health benefit package design in Kazakhstan. Methods: The Ministry of Health of Kazakhstan approved the operationalization and application of EDPs during March 2019 and December 2020. We used a combination of desk research, conducting HTA, online surveys as well as a face-to-face workshop in Nur-Sultan, Kazakhstan, and two online workshops to prioritize 25 selected health technologies. During the latter, we tested two alternative approaches to prioritization: quantitative multicriteria decision analysis (MCDA) and the use of decision rules. Results: For each of the HTA reports, evidence summaries were developed according to the decision criteria (safety, social priority disease, severity of disease, effectiveness, cost-effectiveness, level of evidence, financial risk protection and budget impact). When appraising the evidence, the advisory committee preferred using quantitative MCDA, and only when this would result in any controversy could decision rules be applied. Conclusions: Despite several challenges, including a partial disruption because of the COVID-19 pandemic, implementation of the process will likely play a key role in determining an evidence-informed and transparent health benefit package.

## 1. Introduction

The Republic of Kazakhstan (Kazakhstan) is a land-locked country in central Asia and has transitioned from lower- to upper-middle-income status in 2006. The country is subdivided into 17 administrative divisions. Life expectancy over the past ten years increased to 76.4 years for females, and 67.5 for males (as of 2017). This is lower than the average in OECD countries (over 80 years) [1]. The same applies to the amount of health care expenditure, which is 3.14% per gross domestic product (2015 figures), while out-of-pocket expenditure (OOP) is high at 33.15%.

Like many other countries in the world, Kazakhstan considers health technology assessment (HTA) as an important policy instrument on the path towards achieving universal health coverage (UHC) [2]. HTA is defined as “a multidisciplinary process that uses explicit methods to determine the value of a health technology at different points in its lifecycle. The purpose is to inform decision making in order to promote an equitable, efficient, and high-quality health system [3].” The Republican Center for Health Development (RCHD) holds the responsibility to examine and provide information about the effectiveness, safety, cost-effectiveness, and budgetary impact of devices, diagnostic methods, and other health technologies from the public health perspective of Kazakhstan.

In May 2019, the Minister of Health signed an act that provided the possibility of conducting HTA to inform health benefit packages. From January 2020, the aim was for the population to benefit from the State Guaranteed Benefit Package (SGBP) funded from the government budget. The services under the SGBP were mainly emergency care and outpatient care. Additional services, including inpatient care, would be provided through the Mandatory Social Health Insurance (MSHI) system which was introduced in 2017. Only employers, individual entrepreneurs, private lawyers, court executors and other persons who receive income based on civil contracts were eligible for the MSHI package. Kazakhstan’s MSHI system was aimed at increasing the competitiveness of the healthcare sector and improving the quality of health services.

This paper reports on the process used to determine both the SGBP and the MSHI benefits packages in the period March 2019–December 2020, employing evidence-informed deliberative processes (EDPs).

The EDP framework (see Figure 1) is a practical and stepwise tool for HTA bodies with the explicit aim to optimize the legitimacy of benefit package decisions [4].

The practical guidance on EDPs that was developed by Radboudumc, and of which WO is the lead author, provides recommendations on how these elements can be implemented in each step of the decision-making process of benefits package design. The guide takes the current decision-making context in a country as the starting point, offering practical support depending on the country’s level of HTA development.

We subsequently report on the EDP implementation and each of the Steps, A through D, excluding communication and appeal (Step E) and monitoring and evaluation (Step F). For the latter two steps, we provided future guidance to the MoH. The paper concludes with a discussion of these results from a broader perspective.

## 2. Materials and Methods

The operationalization and application of EDPs for health benefit package design for Kazakhstan was funded by the Ministry of Healthcare of the Republic of Kazakhstan and executed by an international project team (including all authors of the paper) of Radboud university medical center, located in the Netherlands. The project was initiated during a two-day meeting between the project team and representatives of the Ministry of Health, including the acting vice-minister, on 16–18 April 2019 in Nursultan, Kazakhstan. For the operationalization of each step of the EDPs, we used a combination of:desk research to conduct a situational analysis (all steps). This included existing and future relevant regulations, data for monitoring the health-related 2020 Strategic Development Goals in Kazakhstan, WHO Global Health Data and OECD data;questionnaire to two senior HTA representatives of the RCHD, including the acting head of the HTA department, to gain a more in-depth understanding of certain aspects related to the current conduct of HTA in Kazakhstan;online survey on EDPs among 13 representatives of the Ministry of Health (*n* = 6), Social Health Insurance Fund (*n* = 1), and RCHD (*n* = 6) (all steps);online surveys among 15 members of the advisory committee that was formed for this project (step A) to elicit their views regarding coverage decision criteria (step B), selecting health technologies for assessment (step C) and for determining the weights of decision criteria to be used in the quantitative MCDA (step D3);a face-to-face workshop in Nursultan, Kazakhstan on 3 September 2019 with the advisory committee to define decision criteria (step B);work visit to Kazakhstan in the week of 17 February 2020 to engage with the RCHD who are supporting the development of the HTA reports (step D1-D2);conducting 25 HTA reports according to international standards, and weekly video-conference sessions with the assessment team to discuss progress and issues during February–August 2020 (step D1-D2);two online workshops (23 September and 6 October 2020) with the advisory committee to prioritize 25 selected health technologies, using quantitative multicriteria decision analysis or decision rules (step D3).

We will elaborate on the methods when reporting on the EDP implementation step by step below.

## 3. Results

### 3.1. Step A: Installing an Advisory Committee

A survey was conducted among 13 representatives of the MoH, Social Health Insurance Fund, and the RCHD. The survey was developed based on the EDP framework, consisting of elements that reflect each part of the framework and the contextual factors for HTA development and use. Respondents were asked about the presence of each element (three-point scale: present; present to some extent; not present), and whether this needed guidance (yes/no). Furthermore, we asked if any element was missing and for potential examples of best practices relating to each step of the framework. The survey has been used previously to collect information on the use of EDPs by members of the International Network of Agencies for Health Technology Assessment (INAHTA) [5] and experts from low- and middle-income countries [6].

The survey showed that an advisory committee, the Joint Commission on Healthcare Quality (JHCQ), is present in Kazakhstan. This commission is a permanent advisory body of the MoH and is established to develop recommendations for improving clinical protocols, standards for medical education, drug supply, and a system for controlling the quality and accessibility of health services. However, 76% of the respondents felt that guidance was needed for installing an advisory committee. It was mentioned that there are committees with overlapping functions and responsibilities and that the members of the JHCQ and their subcommittees need training in HTA. It was felt that guidance was also needed (92%) in the roles and responsibilities of such a committee, the stakeholders involved in the process, as well as the formal approach followed by the committee.

Several options for installing a committee were proposed to the MoH based on international examples as presented in the EDP guide. The MoH decided on the composition of an advisory committee consisting of 15 permanent members reflecting broad stakeholder groups, with the responsibility of developing recommendations on HTA and benefit package design. The advisory committee comprised the vice-minister of the MoH, who acted as chair, two senior representatives of relevant MoH departments, two senior representatives of the Social Health Insurance Fund, two HTA experts and a scientist with ethics expertise of the RCHD, two senior representatives of public associations, two patient representatives, a clinician (oncologist), a nurse, and an expert in disease management.

### 3.2. Step B: Defining Decision Criteria

A survey to develop a consensus on the importance and definition of criteria for the prioritization of health technologies was developed based on a review of Kazakhstan strategic documents and international best practices. It was distributed electronically to the 15 advisory committee members of which 11 responded. The non-respondents included the two representatives of the MoH, the two representatives of the public associations and one patient representative. The survey was sent prior to the face-to-face workshop (held on 3 September 2019), which was attended by 10 committee members representing all stakeholder groups, and one member delegated the task to a colleague (oncologist). During the workshop it became clear that committee members had a very high level of agreement regarding the decision criteria for health benefit package design. The following criteria were selected by the advisory committee and approved by the MoH: social priorities, financial burden for households (financial risk protection), severity of disease, effectiveness, level of evidence of effectiveness, safety, cost-effectiveness, level of evidence of cost-effectiveness, and budget impact. The description of the criteria was based on international standards, including the HTA glossary [7].

### 3.3. Step C: Selecting Services for Evaluation

One of the tasks of the advisory committee was to compile a list of 25 health technologies that will be subject to full HTA. During the workshop held in Nursultan on 3 September 2019 the advisory committee agreed to use the burden of disease and budget impact criteria for selecting technologies. A local study team member worked with the MoH and subordinate organizations to undertake the following steps in selecting 25 health technologies.

Include all technologies from 2016 to 2018 (89 health technologies) that were recommended by the JHCQ as well as technologies that were prioritized by RCHD but that have not yet have been assessed by the RCHD for 2019 (22 technologies).The entire list of 111 (only new) technologies was classified by the disease type according to the top diseases across burden of disease (in terms of disability-adjusted life years) and mortality in Kazakhstan: ischemic heart disease, stroke, neonatal disorders, respiratory diseases, and cancer. We focused on the technologies that were targeting these diseases, which led to 64 eligible candidates for selection. It appeared that the majority of the 64 health technologies are related to the field of oncology (*n* = 43; 67%).The advisory committee members could propose technologies from the existing list of medical services targeting diseases with a high burden of disease. This exercise led to 19 additional health technologies.The (potential) budget impact of 54 out of the 83 health technologies were estimated (not for all health technologies cost data could be found).The list of 83 *new* health technologies including available information on costs was sent to the advisory committee members for selection. They were asked to make an initial selection on the basis of the highest (potential) budget impact. Members were then asked to choose technologies that target different disease groups. In addition, they were instructed that when considering an *existing* health technology (*n* = 19), they were advised to select those with potential low or no evidence regarding effectiveness, or those that may potentially be excluded from the benefit package, and / or have a potential large budget impact. All advisory committee members responded and the level of agreement between them ranged from 0% to 79%. We felt it was acceptable to select those health technologies with a level of agreement of at least 50%. This was the case for 22 health technologies. We complemented the list with two health technologies that had 43% level of agreement and one existing health technology. See Table 1 for an overview of the selected health technologies across diseases and type of technologies.

**Table 1 ijerph-19-11412-t001:** Distribution of 25 final selected technologies across diseases and type of technologies.

Disease Category	Type of Intervention	Number of Health Technologies
oncology	intervention	4
oncology	device	4
oncology	medicine	4
circulatory system diseases	intervention	1
circulatory system diseases	device	7
circulatory system diseases	medicine	1
Neonatal diseases	medicine	1
Ischemic heart disease	device	1
Diabetes	medicine	1
Not related to the top five health burden	device	1

In determining which health technologies could be considered for reimbursement via the SGBP or MSHI, we proposed that if the health technology targeted a social priority condition listed in the Order of the MoH (dated 17 October 2019 No. ҚP ДCM-13) then it would be considered for inclusion in the SGBP. If not, the health technology could be considered for inclusion in the MSHI. This was agreed by the MoH. The social priority diseases are:TuberculosisDisease caused by human immunodeficiency virus (HIV)Chronic viral hepatitis and cirrhosisMalignant neoplasmsDiabetesMental and behavioral disordersChildren’s cerebral palsyAcute myocardial infarction (first six months)RheumatismSystemic lesions of connective tissueDegenerative diseases of the nervous systemDemyelinating diseases of the central nervous systemOrphan diseases

### 3.4. Step D1–D2: Scoping and Assessment

Before undertaking the assessments, we reviewed the quality of seven existing (‘old’) HTA reports, conducted by the RCHD, covering a sample of priority disease areas and types of technologies (devices, interventions and medicines) using the INAHTA checklist on HTA reporting [8,9]. The quality review of the HTA reports showed great variation in previously developed reports and that all aspects of HTA reports had significant room for improvement. From this review it was recommended that HTA reports should be customized and made relevant for the country context.

The objective was to develop an HTA product defined by INAHTA as a rapid review [10], as this aligned best with the current HTA capacity and resources in Kazakhstan. A standardized template of the HTA report was developed following EUnetHTA standards [11,12] to ensure that all the reports were developed in the same manner, and in line with the MoH Order 18717 (May 2019).

A dedicated assessment team, including three local HTA experts, collected available evidence on all decision criteria, excluding financial risk protection, for all 25 technologies. Literature reviews were conducted to assess the safety, effectiveness and cost-effectiveness. A local team worked with the MoH and subordinate organizations to obtain information on intervention and comparator costs to determine incremental costs. We estimated the incremental cost as information on the cost-effectiveness of health technologies was frequently unavailable. The estimations of the budget impact were presented in actual figures, while for the incremental costs we used a classification scale (less, equal or more expensive).

An evidence summary sheet was developed which allowed for the classification of the decision criteria according to the measurement levels (i.e., high, medium, low) as well as a short description or an extract from the HTA report justifying the classification.

We were not able to retrieve any information for financial risk protection hence this was excluded from the (quantitative) MCDA. Rather it was only referred to in the deliberation where the advisory committee had to provide a (verbal) qualitative indication on the question: “How much will a patient need to pay out of pocket when this intervention will not be (completely) covered?”

### 3.5. Step D3: Appraisal

The MoH agreed to conduct appraisal of the 25 health technologies to be undertaken using two methods: 1) quantitative MCDA and 2) decision rules. This was requested by the advisory committee to demonstrate the differences between the two methods, of which the committee would adopt a single method into their processes going forward.

It was agreed that health technologies eligible for reimbursement under the SGBP (13 technologies) were evaluated using the ***quantitative MCDA method*** and those eligible for reimbursement under the MSHI (12 technologies) were evaluated using ***decision rules***. Health technologies were eligible for reimbursement under the SGBP if they were classified as a social priority and listed in the Order of the MoH (dated October 17, 2019, No. ҚP ДCM-13), if not, they were then eligible under the MSHI.

We defined each criterion and used measurement levels for each criterion using international agreed standards and key references [7,13,14,15,16,17,18,19] (Appendix A). All criteria (except for social priority) were accordingly classified in an evidence summary table as shown in Table 2 for all health technologies. The classification was performed by one researcher and checked by another researcher. In case of disagreement, the researchers discussed the classification with a third researcher. The classification assists with the prioritization of each health technology (i.e., ranking) in both analyses.

### 3.6. Quantitative MCDA Method

Quantitative MCDA uses a value measurement model to interpret the performance matrix, followed by deliberation. To determine the value of a technology for use in the quantitative MCDA, an online survey was distributed to the advisory committee members requesting them to score the following criteria: effectiveness (including quality of the evidence), safety, severity of disease. We instructed them to allocate a total of 100 points between these four criteria by requesting the following: “Please rate the importance of the following decision criteria by distributing 100 points between them to reflect their importance.” We calculated the criteria weights by taking the average of the points provided by the committee members.

Cost-related criteria such as cost-effectiveness and budget impact are also important criteria, however, these should not be part of the quantitative MCDA as it is unrealistic to assume that committee members can adequately attach weights to them as they are likely not aware of health budget constraints and alternative ways of using resources [20]. Therefore, we only considered these later in the MCDA process, in the deliberative part. Furthermore, as the criterion social priorities was used to classify the health technologies to be considered for the SGBP or the MHSI, there was no need to score this criterion as well.

The survey yielded a high response rate (12 out of 15 advisory committee members) for which we were able to assign weight to criteria given (Table 3) for the criteria to be used in the quantitative MCDA.

To undertake quantitative MCDA, we performed the following steps.

Allocate a performance score based on its classification, to the criteria: severity of disease, effectiveness, level of evidence of effectiveness and safety.Multiply scores with criteria weights.Calculate the total MCDA score by adding up the sums for each health technologyRank health technologies based on the total score.Include additional economic information (budget impact analysis and incremental cost) for use in the deliberation step.Undertake a deliberation with the advisory committee to arrive at a consensus on ranking.

Two online meetings were organized with the advisory committee to discuss the ranking of selected health technologies. The committee indicated their understanding of the quantitative MCDA and did not provide any adjustments to the rankings.

### 3.7. Decision Rules

To undertake the analysis, we performed the following steps.

Knock out of health technologies.

The advisory committee members agreed during its meeting in September 2019 to use effectiveness as a first knock-out criterion before considering other criteria-in effect, this is the use of a ‘decision rule’. This also applies to safety as it would be unethical to provide unsafe treatments to patients. However, in our analysis, we did not have any health technologies under evaluation that were ‘not clinically effective’ or ‘not-safe’. Therefore, there were no health technologies that were excluded from this step.

2.Prioritization according to potential cost-effectiveness and severity of disease.

Generally, cost-effectiveness thresholds are set for the local context. Cost-effectiveness analysis then needs to take place to estimate quality-adjusted life years (QALYs) or disability-adjusted life years (DALYs) to determine if it is highly, moderately, or not cost-effective. In our project, the evaluation of cost-effectiveness was based on literature reviews. Given that the majority (all but one) of health technologies did not have any local estimates on cost-effectiveness, we were only able to assume the potential cost-effectiveness based on international estimates. The level of evidence also plays an important factor when interpreting the cost-effectiveness data.

In this step, health technologies were ranked from highest performing according to the combined ranking of their performance in potential cost-effectiveness and severity of disease. The highest-ranked health technologies were those which were rated as highly cost-effective with moderately severe disease (as no interventions had a high severity of disease). The lowest-ranked interventions were those that were not cost-effective (or no-information) and a ‘not-severe’ disease (see Table 4).

3.Include additional information on other relevant decision criteria.

Once the health technologies were ranked according to priority categories (combination of cost-effectiveness and severity of disease), we included further information on economic (LOE of cost-effectiveness, budget impact and incremental costs) and clinical information (effectiveness, LOE of effectiveness, safety). The classification details are similarly given in Table 2.

4.Undertake a deliberation to arrive at a consensus on ranking.

As described earlier, two meetings were organized with the advisory committee. The committee expressed that they understood the ranking using decision rules and did not provide any adjustments to the rankings. However, the advisory committee preferred using quantitative MCDA, and only when this would result in any controversy could decision rules be applied.

## 4. Discussion

Benefit package design is without a doubt intrinsically complex, and there are increasing demands to ensure that health benefit package design is fair and legitimate. By using EDPs, an explicit process is developed ensuring that the format and content are consistent with expectations of the stakeholders involved. Having an explicit process enables more effective collaboration and sharing of information to overcome challenges of variance in the extent and scope of analysis and differences in reporting the results.

Standards for presenting information largely remove authors’ judgements on what, and how, evidence should be reported. The INAHTA checklist does not help in all areas particularly concerning the assessment of content quality, however, it does assist in the examination of what content is presented. In assessing the quality of assessments and reporting, we found an improvement when making use of the INAHTA checklist. HTAs in Kazakhstan have been undertaken on a wide range of topics by a variety of individuals. We expect a growing body of evidence to be generated from HTA for decision making, however, there could be a concern in the variation in the quality of HTA reports and approaches taken in their preparation. By instituting a process where researchers make use of the INAHTA checklist, we expect higher quality and more transparent HTA reports to be produced. Overall, this should contribute to better decisions being made by policymakers.

There were several practical challenges to the implementation of each of the steps of the EDP framework. Firstly, on the *advisory committees* (Step A), a survey indicated that guidance was needed for installing members of the advisory committee and we were successfully able to establish a committee using the principles outlined in the EDPs. A survey also indicated that the members of the advisory committee needed training in HTA. We believe that not all members may have the capacity to fully grasp the presented evidence as there were limited opportunities to train members although we did provide instructions and explanations in all steps, such as criteria explanation sheets, evidence briefs and had explicit rounds for clarification questions during appraisal. The success of this is uncertain as we were unable to measure the impact of these.

Another area of concern are the changes in the leadership within the MoH. At the time of formation, the vice-minister of the MoH was appointed chair of the committee however, there have since been changes in the leadership which makes it challenging for training and results in the loss of institutional memory.

Second, on the *decision criteria* (Step B), burden of disease was initially proposed as a generic decision criterion. However, after discussion with the broader study team, we advised not to use this criterion since using burden of disease would systematically prioritize more common diseases over more rare diseases. Furthermore, burden of disease is being used as a selection criterion for prioritizing HTA. Instead, we recommended the use of severity of disease as a decision criterion. Using severity of disease would inform advisory committee members on the related severity of diseases that health technologies address at the individual level, without systematically giving common diseases more priority over rare diseases. A survey among workshop participants showed that committee members had a very high level of agreement concerning both the use of proposed criteria and definitions.

Third, regarding the *assessment* of interventions (Step D2), the team faced several challenges in compiling evidence. Due to both capacity and time constraints, only a review of high-level evidence and restricted literature searches were conducted. While this is in line with INAHTA rapid review characteristics, there were challenges in the contextualization of evidence and this was not always feasible given the lack of data in the country or region as a whole.

There were also challenges in writing up ‘standardized’ assessment reports as there were numerous researchers, each with different levels of capacity (both in time and skills). While the template indicated the type of information needed, it did not provide information on the level or breadth of information to be included. At times there was too much, or to little information included. Researchers were able to successfully compile completed reports but it was noted that more local evidence would be an improvement to the process. However, we feel that the methods used to assess the evidence were acceptable and would still contribute to improved decision making.

In addition, there were challenges in assessing information regarding some decision criteria. The assessment of financial burden for households and financial risk protection, both measures of equity, was not possible, so they were excluded. This highlights that certain criterion recognized as being important, such as ‘equity’, have then been excluded due to their complexity.

Fourth, in the *appraisal* of interventions (Step D3), several aspects may have compromised the decision-making processes by the advisory committee. It is unclear if all committee members had sufficient background knowledge to fully understand and interpret the presented evidence for all criteria. We did provide descriptions of the evidence, but it is not clear whether this was sufficient, we know from experience that interpretation of evidence can be cognitively challenging.

The health technologies were rank ordered by the assessment team using MCDA methods, however, there were few comments raised by committee members expressing their own judgement on the ordering. The advisory committee preferred using quantitative MCDA as they felt this would be a better fit for determining the health benefit package for Kazakhstan, i.e., it could be used for all health technologies, whether it is a social priority or not. Specifically, they mentioned that quantitative MCDA is more acceptable as it clearly shows a ranking based on the available evidence. Reducing the cognitive load of processing several criteria simultaneously has often been mentioned as an advantage of quantitative MCDA [20]. However, it is increasingly known that MCDA also has limitations and should therefore be critically considered before it is used for benefit package design [21]. There is a tendency of participants in MCDA studies to accept the quantitative outcomes as final results, whereas these should only be considered as a starting point for deliberation in order to consider other (non-quantifiable) criteria [21]. Specifically, as implemented in this study, MCDA studies should not cover costs and cost-effectiveness as criteria in the quantitative analysis, and these should be considered in the deliberative process alongside the MCDA results [21]. However, it was not clear whether committee members adequately took these criteria into account in the deliberative process. Therefore, an important contribution to the process would be to more actively engage stakeholders in deliberations by exchanging their views based on argumentation and evidence. This could be through a process whereby each committee member is required to document their argumentation on paper and then each share his/her argumentation before voting, thereby stimulating a more in-depth discussion. This would provide all committee members maximum influence on the decision-making process. Nevertheless, the members were positive overall about the appraisal process.

Fifth, the overall decision-making process was heavily disrupted by the COVID-19 pandemic. Committee meetings that helped establish the process were held on-site until February 2020, but this transitioned fully online afterwards. This has likely compromised stakeholder participation and the quality of the decision-making process.

## 5. Conclusions

The implementation of the stepwise approach faced several challenges including a partial disruption because of the COVID-19 pandemic, but implementation of the process will likely play a key role in determining an evidence-informed and transparent health benefit package. An important contribution of this paper is explaining how policymakers can improve decision making with limited capacity and resources. Limited capacity can be overcome by training a small team to undertake rapid assessments of existing evidence thereby increasing the number of technologies that are reviewed. The limited resources not only relate to human and financial resources but also the limited amount of country specific data. While there may be limited studies conducted within Kazakhstan, there is a plethora of data that exists internationally which can be used for decision making.

## Figures and Tables

**Figure 1 ijerph-19-11412-f001:**
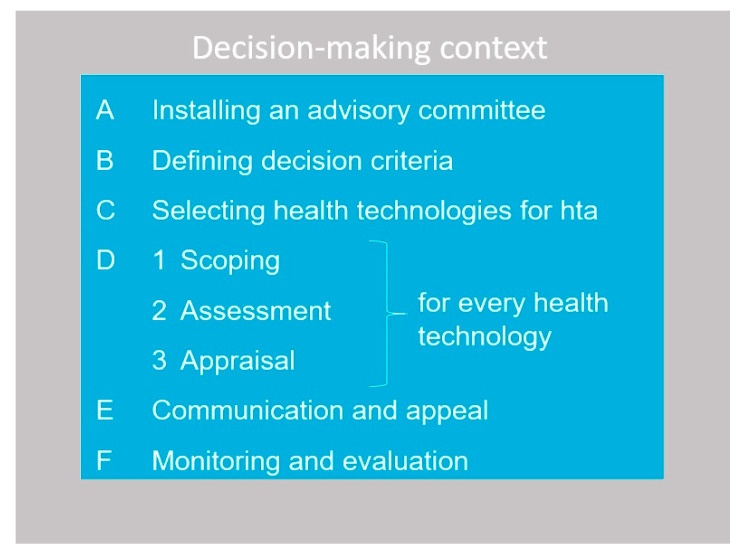
Six steps of implementing EDPs.

**Table 2 ijerph-19-11412-t002:** Classification options.

Criteria	Classification Options			
1 Social priority	Yes, a social priority		No, not a social priority	
2 Severity of disease	1. Severe	2. Moderately severe		3. Not severe
3 Effectiveness	1. Effective (much better than comparator)	2. Comparable effectiveness		3. Not effective (much worse than comparator)
4 LOE *: Effectiveness	1. Very confident	2. Moderately confident		3. Limited confidence
5 Safety	1. Much better than comparator	2. No difference (compared to comparator)		3. Much worse than comparator
6 CE **	1. Highly cost-effective	2. Moderately cost-effective		3. Not cost-effective
7 LOE: CE	1. High level of evidence	2. Moderate level of evidence		3. Low level of evidence
8 Costs	1. Less expensive	2. Equal cost		3. More expensive
9 BI ***	1. Low BI	2. Moderate BI		3. High BI

* LOE = level of evidence; ** CE = cost-effectiveness; *** BI = budget impact.

**Table 3 ijerph-19-11412-t003:** Criteria weights.

Criteria	Weights (Total = 100%) *
Severity of the disease	15.42%
Effectiveness	33.75%
Level of Evidence: Effectiveness	25.42%
Safety	25.42%

Note: * Rounding was used as such it does not add up 100%.

**Table 4 ijerph-19-11412-t004:** Priority category.

Priority Category	Potential Cost-Effectiveness	Severity of Disease
1	1. Highly cost-effective	2. Moderately severe
2	1. Highly cost-effective	3. Not severe
3	2. Moderately cost-effective	2. Moderately severe
4	2. Moderately cost-effective	3. Not severe
5	3. Not cost-effective	2. Moderately severe
6	3. Not cost-effective	3. Not severe
7	No info	2. Moderately severe
8	No info	3. Not severe

## Data Availability

Not applicable.

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
