# Peer review of "The Use of Evidence-Informed Deliberative Processes for Health Benefit Package Design in Kazakhstan"

_ijerph, 2022, doi:10.3390/ijerph191811412_

Round 1
Reviewer 1 Report
Dear authors,
thank you very much for this paper and delighted to discover that Kazakstan is going through this process of prioritisation.
I only a few minor comments:
1. The sample size of the survey is so small that it does not make sense to report on results using percentages. I would mention that number of people instead assuming that all 13 people answered the survey, which is not mentioned.
2. It would instructive for the reader to know about who the interviewees are with more details.
3. Would it be possible to list the social priorities?
4. Page 6: it would be interesting for the reader to objectively qualify all the adjectives "high, moderate, low" used to characterise criteria. I would also suggest you explain how biases were mitigated.
5. Page 6: could you please explain why not all 9 criteria were used for the MCDA (Table 3)? how did you select them?
6. I would suggest you expand the paragraph related to MCDA in the discussion as there is growing evidence on the limitations of MCDA with a lot of it coming from your own institution.
Thank you for this paper.
Reviewer 2 Report
This paper reports on the use of EIDPs to identify and prioritize interventions for inclusion in health benefits packages in Kazakhstan. The authors report on the procedures used, challenges encountered, and lessons learned.
Minor comments:
1. Lines 128-131: I would appreciate more information on the characteristics of the advisory committee members. Which stakeholder groups do they represent? Are patient groups represented? Civil society? Other public representatives? EIDPs are intended to improve the legitimacy of health priority-setting, in part by ensuring that a range of relevant perspectives are included. It’s also interesting to note that only 10-11 of the members participated in the development of decision criteria. Which perspectives were excluded? More information about the advisory committee would allow the reader to assess the degree to which a range of perspectives has been solicited and incorporated. And it may be that, for practical reasons, it was not possible to involve a range of stakeholders in this step, but this trade-off should then be discussed in the paper.
2. 141-144: Does “level of evidence” refer to the degree of uncertainty in the evidence? The quality of the evidence? Both?
3. 156-159: How did you move from 111 to 64 technologies? Was a specific cut-off used in terms of disease burden or mortality?
4. 162: Similarly here, how was “high burden of disease” defined?
5. 164: How was budget impact estimated?
6. 167-174: What does it mean that committee members made selections “according to their opinion”? For 54/83 technologies there was an estimated budget impact, so couldn’t selections from among these have been based on those estimates alone (through the use of some threshold value)? Did committee members use their judgment for only the remaining 29 technologies? It also seems that criteria in addition to burden of disease and budget impact were used for selecting technologies (e.g. a distributive consideration concerning burden across disease groups and level of evidence). If the authors agree, these additional criteria should be explicitly listed in lines 148-150 as well.
7. Could you please provide some more details on how the criteria weights were determined?
8. 289-291: Why did the committee prefer using quantitative MCDA over decision rules? I would be interested in more detail here. What did they see as the trade-offs between the two?
9. In general, it would be helpful to see more of the work/documents that went into the full process (perhaps in an Appendix). A number of surveys, etc. are mentioned.
